# Everyone Deserves Recourse: Feasible Recourse Paths Using Data Augmentation

## Abstract

Decisions made using machine learning models can negatively impact individuals in critical applications such as healthcare and finance by denying essential services or access to opportunity. *Algorithmic recourse* supplements a negative AI decision by providing rejected individuals with advice on the changes they can make to their profiles, so that they may eventually achieve the desired outcome. Most existing recourse methods provide single-step changes by using counterfactual explanations. These counterfactual explanations are computed assuming a fixed (not learned) distance function. Further, few works consider providing more realistic multi-step changes in the form of recourse paths. However, such methods may fail to provide any recourse path for some individuals or provide paths that might not be feasible, since intermediate steps needed to reach the counterfactual explanation may not be realizable. We introduce a framework for learning an optimal distance function and threshold to compute multi-step recourse paths for all. First, we formalize the problem of finding multi-step recourse paths. Given a set of feasible transitions, we propose a data-driven framework for learning the optimal distance and threshold for each step with PAC (Probably Approximately Correct) guarantees. Finally, we provide a data augmentation algorithm to ensure that a solution exists for all individuals. Experiments on several datasets show that the proposed method learns feasible recourse paths for all individuals.

## 1 Introduction

Machine learning (ML) models are increasingly being used for algorithmic decision-making in high stakes applications. Hence, when individuals are adversely affected by these decisions, the provision of transparent explanations for the negative decisions becomes paramount. For example, consider the scenario where credit line applications of bank customers get denied. The imperative for transparency and explainability is further underscored by regulatory mandates such as the Equal Credit Opportunity Act (ECOA), the Fair Credit Reporting Act (FCRA) (Ammermann, 2013), the 'Right to Explanation' enshrined in the EU General Data Protection Regulation (EU-GDPR) (Goodman & Flaxman, 2017), and the U.S. AI Bill of Rights (House, 2022).

These explanations often take the form of sequential steps aimed at achieving desired or favorable outcomes for affected users. Such recommended steps represent algorithmic recourse, and provide users with a pathway to address adverse decisions by gradually changing their profile to one that most likely receives the positive decision. Single-step recourses frequently rely on counterfactual explanations (CFEs), which propose changes to the input data that would lead to a different decision outcome (Wachter et al., 2017). However, recent research has highlighted the limitations of single-step recourses, advocating instead for multi-step recourse paths towards favorable outcomes (Verma et al., 2020; Venkatasubramanian & Alfano, 2020). It is imperative that such recourse paths remain realistic, meaning they should be both feasible and actionable (Poyiadzi et al., 2020b), in order to effectively assist end-users. Furthermore, algorithms designed to provide realistic recourse paths should be able to provide recourse for every individual i.e., realism constraints should not come at the cost of no recourse for some individuals.

Figure 1 provides a demonstrative example for the need for multi-step recourse paths for all. Individuals represented in red are assigned the negative outcome by a given machine learning loan classifier, and recourse paths need to be found for them. Finding counterfactual explanations and

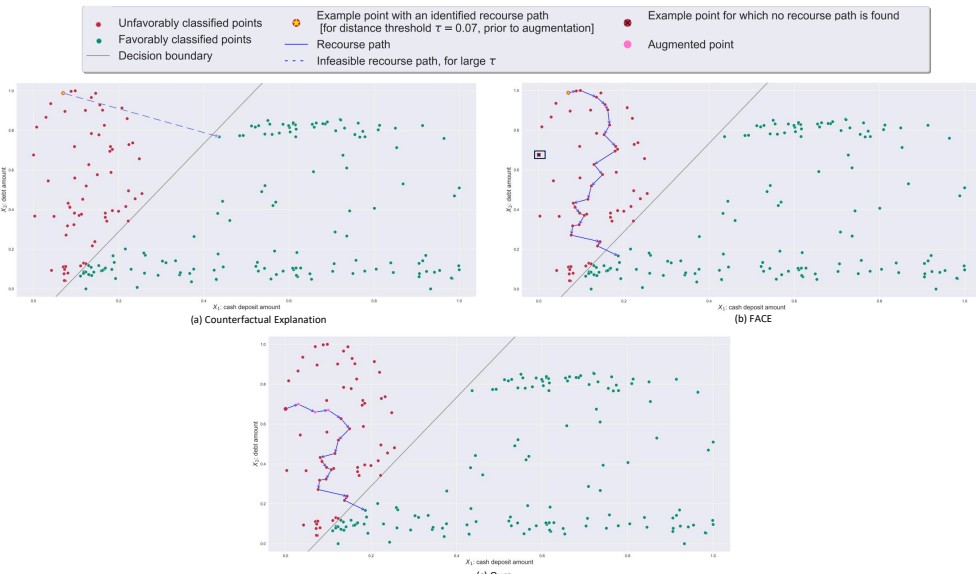

Figure 1: Illustration of a loan classification task with two features: cash deposit amount ($X_1$) and debt amount ($X_2$), with normalized values for each feature. The three plots show three different methods to provide recourse. **(a)**: An example of providing a counterfactual explanation. While a recourse path is found, the recommendation provided can be infeasible since the change in cash deposit amount is huge. **(b)**: An example of a success and a failure of a path-based algorithm. Given a particular value of the threshold $\tau = 0.07$ on the distance between the original point and the next point in the path, connecting points that have distance within this threshold can help find a recourse path for the starred point, but not for the boxed point (the threshold is too low). If, on the other hand, a high value of threshold is chosen (or equivalently a counterfactual explanation is found), then more points may receive recourse, but such recourse is likely to be infeasible. **(c)**: Our method learns the optimal distance and threshold, and then augments to allow for recourse paths to be constructed for points that did not receive a recourse path.

suggesting the line joining these as the path may lead to infeasible transitions, especially when the distance between the input and counterfactual is large (dashed line in 1(a)). In the example shown, the selected individual requires changing their cash deposit amount exorbitantly. Another approach can be to define paths based on consecutively finding nearby individuals in the training set, and moving in the direction of the boundary (Poyiadzi et al., 2020b; Pentyala et al., 2023), as shown in 1(b). However, these methods depend on a predefined distance function and a threshold parameter, which are not learned. If the distance between two points exceeds the threshold, then a transition is infeasible. Setting this distance threshold to a large value is equivalent to finding a nearest neighbor counterfactual. Setting this threshold to a small value results in no recourse for some individuals (boxed point in 1(b)), since it might be impossible to keep expanding the path by finding nearby points within the distance threshold. This is especially problematic in settings such as the loan classification task, where in practice such method would not provide any recourse to some individuals.

This work introduces a distance function and threshold learning framework, combined with an augmentation technique to provide feasible recourse paths for every individual. The method is model-agnostic and only requires access to a classifier's prediction probability output. Given a set of feasible transitions, we are able to learn a near-optimal distance function and threshold, that closely approximate the true feasibility relationship for transitions. Using this learned feasibility relationship, we provide a data augmentation technique that creates a recourse path for every individual that initially received the negative outcome. This is shown in 1(c). The key characteristics of such a path are **1)** all intermediate transitions of it are more feasible and **2)** the final point in it receives the positive outcome of the given classifier. Figure 1 (b) shows a path based on our approach.

Our paper is structured as follows: in Section 2.2 we address the problem of learning feasible transitions. For this problem, we give a hypothesis class containing distance functions and thresholds, for which we prove PAC learnability; we show a bounded VC-dimension and an efficient Empirical Risk Minimization (ERM) algorithm. In Section 2, we propose an augmentation algorithm that can

provide a feasible recourse path for every negatively affected individual and provide convergence guarantees under certain assumptions. Finally, Section 3 contains our experiments on one synthetic dataset and three real datasets. Experiments demonstrate that our method can efficiently provide feasible recourse paths for all. To the best of our knowledge, this is the first work addressing the learning of distance functions and thresholds in a multi-step recourse path setting. We also provide a discussion section on how our method fits in the recourse literature.

## 1.1 BACKGROUND AND RELATED WORK

For a literature overview, (Karimi et al., 2022) provide various definitions, formulations, and solutions to recourse, and highlight connections to other challenges like security, privacy, and fairness. Here, we discuss the methods most relevant to ours.

**Counterfactual explanations as recourse:** A common approach in literature is to treat counterfactual explanations (also known as contrastive explanations) as recourse-by-example (Wachter et al., 2017; Sokol et al., 2019; Ustun et al., 2019; Guidotti, 2022). Yet, recent work has highlighted the limitations of counterfactual explanations for algorithmic recourse, emphasizing the need to consider causal relationships between features (Liu et al., 2024; Bynum et al., 2024).

The counterfactual-based approach when considered in its simplest form (identifying a minimally distant counterfactual) relies on strong assumptions. Indeed, consider an individual who is subject to an unfavorable AI prediction and who is given an example of another individual (real or synthetic) as a 'successful counterpart' and a blueprint for future improvement. For such a recourse to be helpful or reliable, one must assume that the suggested counterfactual data point is plausibly a "future version" of the original rejected individual. This assumption is not generally true, as the algorithms that identify a counterfactual typically only enforce *proximity and sparsity* between the counterfactual and the original point. These methods neither specify the concrete actions needed to attain the counterfactual state, nor do they consider causal process that governs how features change *jointly* over time. To assert that a counterfactual is likely or plausibly achievable for the original point, one solution is to use causal modeling, but as we detail later, this is difficult in real settings. Another approach is to rely on additional safeguards to ensure that a counterfactual might be realistically achievable. Path-based methods, which our solution is an example of, aim to provide such safeguards.

**Causality-aware recourse:** How can we make sure that a given positively classified point (i.e., a counterfactual) represents an achievable state for a given initial point? The most principled approach is to deploy causal reasoning. Yet, methods that take this route have to grapple with prohibitive complexity of accurately modeling causal relationships on multivariate data.

In this line of work, experiments are typically limited to synthetic data on a few variables, suggesting limited applicability of these methods in critical real-life settings (König et al., 2021; Dominguez-Olmedo et al., 2022; Karimi et al., 2020; Dominguez-Olmedo et al., 2023). Furthermore, deploying the machinery of do-calculus (Pearl, 2009) requires starting with a proposed *intervention*, and hence the methods for causally aware recourse must comb through large sets of possible changes that a person might enact, in order to then generate 'feasible' samples from each of the counterfactual distributions for each intervention. This is not only computationally expensive, but also non-intuitive, as ideally we would like to check feasibility of an identified target point, and not the other way around.

A different approach focused on providing recourse for any differentiable machine learning-based decision-making system is in (Joshi et al., 2019). The method relies on modeling the underlying data distribution or manifold, but is not applicable to gradient-free models, which are commonly used in critical settings such as healthcare and personal finance.

**Path-based recourse:** In its minimal form, providing recourse might simply mean identifying a *target point* corresponding to a more favorable AI decision. To find a better target, several methods rely on building *paths* – sequences of point transitions between the initial point and one of its counterfactual points. This is known as *path-based recourse*. The existence of a path grants credibility to the end point of the path as a *feasible* target state. Indeed, it is more likely that a target state is achievable if we can describe a series of small (and hence arguably feasible) steps that lead to the target state.

Path-based approach to identifying recourse involves iteratively optimizing changes to an individual's features, while controlling the radius to which each change is constrained to ensure that each

transition is feasible. (Hamer et al., 2023) introduce Stepwise Explainable Paths (StEP), a data-driven framework offering users interventions to alter outcomes, with privacy and robustness guarantees. A prominent example of a path-based recourse algorithm is FACE (Poyiadzi et al., 2020a) – a model-agnostic method for generating counterfactual explanations. Similar work includes (Small et al., 2023; Nguyen et al., 2023). Despite its many advantages, existing path-based methods suffer from two key shortcomings: a) if the distance threshold used to constrain the search for each 'next step' is too large, recourse path may involve transitions that are either costly, unlikely, or infeasible and b) if the distance threshold is small, the algorithm may fail to find a path for some individuals

The strength of our work is in addressing the shortcomings of existing path-based methods. Mitigating failure to provide recourse is of particular importance, as such failures potentially trigger fairness concerns. Consider data collected under racial or gender-sensitive selection bias. If members of a certain demographic group are under-represented in either data class, it is likely that fewer group members will receive a path to recourse, as FACE and similar algorithms rely on density to identify suitable transitions. In this work, we take a position in asserting that *everyone deserves recourse*.

## 2 ALGORITHMS AND THEORETICAL RESULTS

### 2.1 FORMAL PROBLEM DEFINITION

In our recourse setting, individuals are represented by feature vectors from some feature space $\mathcal{I}$. We are given a set of individuals $V \subseteq \mathcal{I}$, and a trained classifier $f : \mathcal{I} \mapsto [0, 1]$ together with some threshold $\alpha \in (0, 1)$. An individual $x \in \mathcal{I}$ receives the positive outcome of the classifier iff $f(x) \geq \alpha$, and the negative outcome iff $f(x) < \alpha$. We define $V_p = \{x \in V \mid f(x) \geq \alpha\}$ and $V_n = \{x \in V \mid f(x) < \alpha\}$ to be the positive and negative individuals of the input set $V$ respectively.

The goal of our problem is to provide actionable recourse to individuals in $V_n$. For an individual $x$, we define actionable recourse as a path of feasible transitions $x = x_1 \mapsto x_2 \mapsto x_3 \mapsto \ldots \mapsto x_k$, where $x_i \in \mathcal{I}$ for all $i \in [k]$, the transition $x_i \mapsto x_{i+1}$ is feasible and relatively "easy", and $x_k$ is the first positive profile ($f(x_k) \geq \alpha$) in the sequence. The recourse interpretation of the above path is that the individual can gradually change their profile, starting from $x$, and consecutively making the feasible change from $x_i \mapsto x_{i+1}$ for all $i \in [k-1]$, they can reach a positive profile $x_k$.

We assume that we are given a "distance" function $d : \mathcal{I}^2 \mapsto \mathbb{R}_{\geq 0}$ and a threshold $\tau \geq 0$. Then, $x \mapsto y$ is feasible iff $d(x, y) \leq \tau$, under the interpretation that the larger $d$ is the more dissimilar the two individuals that are compared. Note that we do not require $d$ to be a metric. For example, when $d$ is not symmetric we capture directional feasibility; $x \mapsto y$ might be feasible while $y \mapsto x$ is not. To capture how easy a feasible transition is, we assume that we are given a weight function $w : \mathcal{I}^2 \mapsto \mathbb{R}_{\geq 0}$. For a feasible $x \mapsto y$, the larger $w(x, y)$ is the more difficult the transition. This work addresses two problems:

**The Problem of Learning** $(d, \tau)$**:** Prior works assume explicit knowledge of $d$ and $\tau$ to define feasible transitions. In our work, we show how we can explicitly learn $d, \tau$ so that we approximate the ground-truth transition feasibility function as optimally as possible.

**The Augmentation Problem:** Finding feasible recourse for $x \in V_n$ corresponds to finding a path $P_x = \{x = x_1, x_2, x_3, \ldots, x_k\}$ for some $k \geq 1$, such that $x_i \in V$ for all $i \in [k]$, $d(x_i, x_{i+1}) \leq \tau$ for all $i \in [k-1]$ and $y_k \in V_p$. **However, this might not always be possible**. For that reason, we want to augment $V$ by adding a new set of individuals $U \subseteq \mathcal{I}$, such that it is always possible to find a path $P_x = \{x = x_1, x_2, x_3, \ldots, x_k\}$ for some $k \geq 1$, with $x_i \in V \cup U$ for all $i \in [k]$, $d(x_i, x_{i+1}) \leq \tau$ for all $i \in [k-1]$ and $y_k \in V_p \cup U_p$, where $U_p$ are the individuals of $U$ receiving the positive classifier label. We also want the weights $w(x_i, x_{i+1})$ for consecutive profiles to be as small as possible.

### 2.2 LEARNING FEASIBLE TRANSITIONS

Let $h^* : \mathcal{I}^2 \mapsto \{0, 1\}$ be the ground truth function that determines feasibility of transitions, i.e., for any $x, y \in \mathcal{I}$ we have $h^*(x, y) = 1$ if $x \mapsto y$ is feasible, 0 otherwise. Let

$$\mathcal{H} = \left\{ h_{d,\tau} = \mathbb{1}_{\left\{ (x,y) \in \mathcal{I}^2 \mid d(x,y) \leq \tau \right\}} \mid d \in \mathcal{D} \text{ and } \tau \in \mathbb{R}_{\geq 0} \right\} \quad (1)$$

---

**Algorithm 1** Computing the ERM classifier $\bar{h}$ for bounded $\mathcal{D}$

---
**Input:** $\mathcal{D}$ and $S = \{(x_i, y_i, h^*(x_i, y_i)) \mid i \in [m]\}$.

1: $ml \leftarrow \infty$
2: **for** each $d \in \mathcal{D}$ and each $\tau \in \{d(x_i, y_i) \mid i \in [m]\}$ **do**
3:    **if** $L_S(h_{d,\tau}) < ml$ **then**
4:      $\bar{h} \leftarrow h_{d,\tau}, ml \leftarrow L_S(\bar{h})$
5:    **end if**
6: **end for**

---

where $\mathcal{D}$ is just a set of "distance" functions from $\mathcal{I}^2$ to $\mathbb{R}_{\geq 0}$. Given $\mathcal{D}$ as the set of "distance" functions we are interested in, $\mathcal{H}$ is a hypothesis class, whose individual hypotheses are parameterized by $d$ (the specific distance function they use) and $\tau$ (a threshold). Then, each such hypothesis $h_{d,\tau}$ returns 1 for supposedly feasible transitions and 0 otherwise, and is of the form

$$h_{d,\tau}(x,y) = \begin{cases} 1 & \text{if } d(x,y) \leq \tau \\ 0 & \text{otherwise} \end{cases}$$

For any $h \in \mathcal{H}$ and $(x,y) \in \mathcal{I}^2$, let $\ell(h,x,y)$ be the $0-1$ loss function, i.e.

$$\ell(h,x,y) = \begin{cases} 1 & \text{if } h(x,y) \neq h^*(x,y) \\ 0 & \text{otherwise} \end{cases}$$

Also, let $L(h)$ be the expected loss of $h$, where the expectation is over randomly drawing two individuals $x, y$ from $\mathcal{I}$ according to the data producing distribution; $L(h)$ can be viewed as the real loss of $h$. In addition, for a training set $S$ that contains $m$ labeled i.i.d. sampled pairs $(x^i, y^i, h^*(x^i, y^i))$, we define the empirical loss of classifier $h$ as $L_S(h) = \sum_{i=1}^{m} \ell(h, x^i, y^i)$.

We want to choose $d \in \mathcal{D}$ and $\tau$ such that $L(h_{d,\tau})$ is as small as possible. Let $\bar{h} = \arg\min_{h \in \mathcal{H}} L_S(h)$ be the empirical risk minimizer (ERM) of $S$. We call $\rho$-ERM, with $\rho \geq 1$, a $\tilde{h} \in \mathcal{H}$ such that $L_S(\tilde{h}) \leq \rho \cdot L_S((\bar{h}))$. Towards our goal, we use the following fundamental theorem.

**Theorem 2.1** ((Shalev-Shwartz & Ben-David, 2014)). *Let $\mathcal{H}$ be a hypothesis class, and let $\epsilon, \delta \in (0,1)$ be any desired accuracy and confidence parameters respectively. Let $VC$ be the VC-dimension of $\mathcal{H}$[1]. Let $S$ be a training set with at least $O(\frac{VC + \log \frac{1}{\delta}}{\epsilon^2})$ training examples and $\tilde{h}$ a $\rho$-ERM of $S$. Then, with probability at least $1 - \delta$ we have $L(\bar{h}) \leq \rho \cdot \min_{h \in \mathcal{H}} L(h) + O(\epsilon)$.*

What the above theorem says, is that the $\rho$-ERM of the training set $S$ approximates well the best hypothesis of $\mathcal{H}$ with high probability, provided that the training set is large enough. For this theorem to be applied, $VC$ needs to be bounded. For a definition of VC-dimension see (Shalev-Shwartz & Ben-David, 2014). In what follows we show a couple of examples of hypothesis classes with bounded VC-dimension, where a $\rho$-ERM is also efficiently computable.

### 2.2.1 THE CASE OF BOUNDED $\mathcal{D}$

We first prove that for our hypothesis class as defined in (1), $VC$ is bounded as long as $|\mathcal{D}|$ is bounded (user-defined fixed and finite set $\mathcal{D}$). Specifically, $VC$ depends on $|\mathcal{D}|$ in an inverse exponential way, which makes the required sample complexity highly practical in $|\mathcal{D}|$.

**Theorem 2.2.** *Let $VC$ be the VC-dimension of the hypothesis defined in (1). If $|\mathcal{D}|$ is bounded, let $N$ be the smallest integer such that $|\mathcal{D}| < \frac{2^N}{N+1}$. Then, $VC \leq N$.*

We now show that the ERM classifier $\bar{h}$ can be computed efficiently in this case (we get an $\rho$-ERM with $\rho = 1$). Details are provided in Algorithm 1.

**Theorem 2.3.** *The classifier $\bar{h}$ computed by Algorithm 1 is an ERM when $\mathcal{D}$ is bounded.*

Combining theorems 2.1, 2.2 and 2.3 proves the following, which essentially says that an accurate feasibility classifier can be computed efficiently for the bounded $\mathcal{D}$ case.

---
[1]The VC-dimension is a measure of learning complexity for a hypothesis class

**Algorithm 2** Augmentation Algorithm

**Input:** Sets $V, V_p, V_n$, functions $d, f, w$, parameters $\alpha, \tau, \lambda$.

1: **for** each $x \in V_n$ **do**
2:     $U \leftarrow \emptyset$
3:     Initialize recourse path $P_x \leftarrow [x]$.
4:     **while** True **do**
5:       Let $x'$ be the end point of the path $P_x$.
6:       $q \leftarrow \arg\max_{y \in V \cup U} \left\{ \frac{\lambda}{w(x',y)} + (f(y) - f(x')) \right\}$
7:       **if** $d(x', q) > \tau$ or $q \in P_x$ **then**
8:         $q \leftarrow \arg\max_{\substack{y \in \mathcal{I} \setminus (U \cup V) \text{ s.t.} \\ d(x',y) \leq \tau}} \left\{ \frac{\lambda}{w(x',y)} + (f(y) - f(x')) \right\}$ and $U \leftarrow U \cup \{q\}$
9:       **end if**
10:      Extend $P_x$ by appending $w$ as its new end point.
11:      **if** $f(q) \geq \alpha$ **then**
12:        Save $P_x$ as the recourse path of $x$ and break the while loop.
13:      **end if**
14:     **end while**
15: **end for**

**Theorem 2.4.** *Let $\mathcal{H}$ be the hypothesis class defined in (1) with bounded $\mathcal{D}$, and let $\epsilon, \delta \in (0,1)$ be any desired accuracy and confidence parameters respectively. Let $N$ be as defined in Theorem 2.2. Let $S$ be a training set with at least $O(\frac{N + \log \frac{1}{\delta}}{\epsilon^2})$ training examples. Then $\bar{h}$ can be efficiently computed, and with probability at least $1 - \delta$ we have $L(\bar{h}) \leq \min_{h \in \mathcal{H}} L(h) + O(\epsilon)$.*

### 2.2.2 THE CASE OF A MORE STRUCTURED AND UNBOUNDED $\mathcal{D}$

We now study a more flexible scenario, where $\mathcal{D}$ does not need to be finite. Here we assume that each $x \in \mathcal{I}$ has $n$ features. We use $\mathcal{I}_j$ to denote the domain of feature $j \in [n]$. For every feature $j \in [n]$, let $f_j : \mathcal{I}_j^2 \mapsto \mathbb{R}_{\geq 0}$ be a given comparison function for that feature. In other words, $f_j(a, b)$ captures the difficulty in changing feature $j$ from value $a \in \mathcal{I}_j$ to value $b \in \mathcal{I}_j$; the higher $f_j(a, b)$ is the more difficult/improbable the change. Let also $g : \mathbb{R}_{\geq 0} \mapsto \mathbb{R}_{\geq 0}$ be a given strictly monotonically increasing function. For any $\beta \in \mathbb{R}_{\geq 0}^n$, we define the similarity/comparison function:

$$d_w(x, y) = g\Big( \sum_{j=1}^n \beta_j \cdot f_j(x_j, y_j) \Big), \quad \text{for } x, y \in \mathcal{I} \tag{2}$$

Note that this form of distance/comparison function encapsulates a plethora of widely used distance functions, such as (weighted) LP-norms. Now, the set $\mathcal{D}$ from (1) will contain all functions that can be defined by any $\beta \in \mathbb{R}_{\geq 0}^n$, and learning a function from $\mathcal{H}$ is equivalent to learning $\beta$ and $\tau$.

**Theorem 2.5.** *The VC-dimension of $\mathcal{H}$ is at most $2n + 1$.*

**Theorem 2.6.** *An $O(1)$-ERM can be computed efficiently, when $\mathcal{D}$ contains all functions of form (2).*

Combining theorems 2.1, 2.5 and 2.6 proves the following; an approximately accurate feasibility classifier can be computed efficiently when distance functions are defined as in (2).

**Theorem 2.7.** *Let $\mathcal{H}$ be the hypothesis class defined in (1) with $\mathcal{D}$ containing all functions of the form 2, and let $\epsilon, \delta \in (0,1)$ be any desired accuracy and confidence parameters respectively. Let $S$ be a training set with at least $O(\frac{n + \log \frac{1}{\delta}}{\epsilon^2})$ training examples. Then a hypothesis $\tilde{h}$ can be efficiently computed, such that with probability at least $1 - \delta$ we have $L(\tilde{h}) \leq O(1) \cdot \min_{h \in \mathcal{H}} L(h) + O(\epsilon)$.*

### 2.3 THE AUGMENTATION ALGORITHM

The algorithm begins by sequentially considering each point of $V_n$. For each $x \in V_n$ it tries to construct a path to some positive point by iteratively expanding the end of the path. Initially, the path is just $x$. If the already constructed path is from $x$ to $x'$ ($x'$ being the end point) then we try to expand

as follows. At first, we look to see if there is a point in the current set of available points $V \cup U$ that can serve as a feasible and easy transition from $x'$, while encouraging this point to be closer to the boundary [2]. Hence, we solve the next optimization problem:

$$q = \arg\max_{y \in V \cup U} \left\{ \frac{\lambda}{w(x', y)} + \big(f(y) - f(x')\big) \right\}$$

If $d(x', q) \leq \tau$ and $q$ is has not been visited before in the path, we expand the path by adding $q$ as the new end point. If this is not the case then we solve the slightly different optimization problem shown below, which tries to find a feasible transition to a new $q \notin V \cup U$, and then we augment $U$ with it.

$$q = \arg\max_{\substack{y \in \mathcal{I} \setminus (V \cup U) \text{ s.t.} \\ d(x', y) \leq \tau}} \left\{ \frac{\lambda}{w(x', y)} + \big(f(y) - f(x')\big) \right\}$$

The first term, i.e., $\lambda / w(x', y)$, guides the optimizer towards choosing transitions that are easy (have small weight $w(x', y)$). In addition, $\lambda > 0$ is a hyperparameter that controls how important this term should be. The second term, i.e., $f(y) - f(x')$, forces the optimizer to move closer to the decision boundary of the classifier $f$, by maximizing the difference between $f(y)$ and $f(x')$; the higher the $f$ value of a point is the closer it is to receiving the positive outcome. At last, the reason we decided to first look for an extension point in $U \cup V$ instead of looking for a fresh point, is because we want to utilize the given dataset as much as possible, since solving the second optimization problem is more time consuming and might give very realistic feature profiles.

On a high level, including $f(y) - f(x')$ in the maximization problem is exactly what helps the algorithm converge. In the ideal case, the points of the path should consecutively move closer to the decision boundary; the $f$ values should consecutively increase along the path, until we finally hit the classification threshold $\alpha$. This could very well be the case in the absence of the first term. However, the presence of the first term might lead some iterations of the algorithm to prioritize small weights in the chosen transitions. By carefully tuning $\lambda$ in our experiments we make sure that the algorithm will converge almost always, even if we have iterations where the $f$-value of consecutive points decreases. Hence, recourse is achieved for everyone. The full details of our approach are in Algorithm 2.

Finally, we provide a formal convergence scenario for our algorithm.

**Theorem 2.8.** *When for all $x \in \mathcal{I}$, there exists $y \in \mathcal{I} \setminus V$ s.t. $f(y) - f(x) > \frac{\lambda}{\min_{a,b} w(a,b)}$, and the algorithm only chooses fresh points (not in $U \cup V$) to expand paths, the algorithm always converges.*

The proof is provided in the appendix.

## 3 EXPERIMENTAL EVALUATION

**Datasets and models:** We evaluate our method on one synthetic dataset generated using a causal graph, and 3 real-world datasets. For details on how we generate the synthetic data see Appendix A. The first real dataset we consider is PIMA (Smith et al., 1988), where the task is to predict if a patient is diabetic. For predictors/features we use glucose levels, BMI, blood pressure, and insulin . The second real dataset is UCI Adult (Becker & Kohavi, 1996), where the goal to evaluate if an individual's income is greater than or less than 50k. For predictors in Adult we use age, education, capital gain, capital loss, and hours per week. Finally, we consider the HELOC dataset (Explainable Machine Learning Challenge), where the goal is to evaluate the risk of performance for credit. We choose external risk estimate, months since oldest trade, average months in file, revolving balance divided by credit limit, and installment balance divided by original loan amount as predictors. For each of the datasets, we train logistic regression models (note that our method is model agnostic and only requires access to prediction probabilities). Additional experiments on two more model classes (gradient boosting and neural networks) are provided in the appendix. Models are trained and all experiments are run by first transforming the data using the minimum and maximum scalar transformation. Details on model performance on the train and test set are provided in the appendix.

**Feasible transitions data:** For each dataset, a set of samples are chosen, and we generate labels for whether a transition is feasible between each of these points using defined rules. Note that these rules

---

[2]This is encouraged but not enforced because some transitions might require moving away from the boundary first. For example, becoming a student may reduce your income but eventually lead to a higher income

are varied to show the effectiveness of our approach. For details on how we generate the labels for the synthetic dataset see Appendix A. For PIMA, we take the entire training set, and for each pair of points label a transition as feasible if the L1 distance for each feature is below the standard deviation for that feature. For Adult, we consider a thousand random samples, and consider transitions to be feasible only if age, education and hours-per-week are increasing and capital gain and capital loss are within a fifth of the standard deviation of their values in the data set. For HELOC we also take 1000 random samples, and we consider transitions to be feasible if the L1 distance for each feature is below the standard deviation for it, the first three features are increasing and the last two are decreasing. The % of the 1 label i.e., transition is feasible, in the pairs created for each set was **1)** $11.31\%$ for the Synthetic Data, **2)** $37.65\%$ for PIMA, **3)** $17.23\%$ for Adult and **4)** for $0.444\%$ HELOC. We discuss the creation and availability of this data in the discussion section.

**Learning $d$, $\tau$:** In all datasets the set $\mathcal{D}$ from (1) contains 5 distance functions: L1, L2, Mahalanobis, Cosine distance, and the Jensen-Shannon distance. We discuss the choice of these distance functions in the discussion section. For Adult and HELOC, the functions also incorporate the natural monotonicity constraints, since such constraints are intuitive. The way we implement this is by returning $\infty$ if the monotonicity is violated. Finally, in all datasets we uniformly subsample 25% of the pairs described in the previous paragraph as the training set of ERM.

**Transition weights:** For a transition between two individuals $x, y$ we use the same weight function as in (Poyiadzi et al., 2020a). Let $f_\rho : \mathcal{I} \mapsto \mathbb{R}_{\geq 0}$ be a likelihood function that depends on the dataset density $\rho$. Note that $\rho$ is computed without the augmented points. Then the weight of the transition from $x$ to $y$ is defined to be $w(x, y) = \frac{d(x,y)}{f_\rho\left(\frac{x+y}{2}\right)}$, where $d$ is the function chosen by ERM.

**Augmentation Solver:** To solve the optimization problem presented in Algorithm 2, the learned distance functions and thresholds are used as constraints and Bayesian optimization is used. The implementation has been taken from (Nogueira, 2014–). To allow for any arbitrary constraints and machine learning models (eg., non-convex decision boundaries), any metaheuristic optimization algorithm can be used. Bayesian optimization is widely used and does not assume any functional form on the target function, and is hence valuable for our problem. In each iteration, the evaluated target function is line 6 from Algorithm 2. Number of iterations and number of initial points are found using grid search. More details are provided in the supplementary material.

**Evaluation of Recourse Paths:** In order to evaluate each recourse method that we test, we use three metrics. At first, let $V_n$ be the individuals that initially received the negative outcome of the classifier, and let $d$ and $\tau$ be the distance function and the threshold respectively, as computed by the ERM.

1. **Validity score:** For every $x \in V_n$, let $P_x$ be its recourse path. Note that in the case of counterfactual explanations $|P_x| = 2$ ($x$ and the computed counterfactual), and when using FACE (Poyiadzi et al., 2020a) we might even have $P_x = \emptyset$. We define the validity $v(P_x)$ of $P_x$ as an indicator variable that is 1 if $P_x \neq \emptyset$ and $d(z, w) \leq \tau$ for every consecutive $z, w \in P_x$, 0 otherwise. The validity score is then defined as validity $VAL = \frac{1}{|V_n|} \sum_{x \in V_x} v(P_x)$.

2. **Average Path Distance and Weight:** For any recourse path $P = \{x_1, x_2, \ldots, x_k\}$, we define the average path distance and weight as $d(P) = \frac{1}{k} \sum_{i=1}^{k-1} d(x_i, x_{i+1})$ and $w(P) = \frac{1}{k} \sum_{i=1}^{k-1} w(x_i, x_{i+1})$ respectively.

**Comparison baselines:** (Pawelczyk et al., 2021) offer a list of implementations for recourse methods. However, most of them only provide single-step recourse paths. We compare to the only implementation that offers multi-step recourse paths (Poyiadzi et al., 2020b), and also compare to finding the nearest neighbor on the other side of the boundary (single -step) for completeness.

## 3.1 RESULTS

**Results from distance function and threshold learning:** In Table 1 we report the outcomes of the ERM algorithm. Since ERM is inherently randomized (recall that we subsample 25% of the given pairs) we ran the algorithm 5 times. In the table we report the mean and the standard deviation of all important metrics across those 5 runs. The metrics under consideration are **1)** the computed threshold, and **2)** the $0 - 1$ error of the resulting feasibility function across the whole set of labeled pairs. We also note that in every dataset, the chosen distance was consistent across all 5 runs.

Table 1: Outcome of ERM on the 4 datasets

|  | Learned Distance | Mean Threshold | SD of Threshold | Mean Loss % | SD of Loss % |
| --- | --- | --- | --- | --- | --- |
| Synthetic Data | L2 | 0.298456 | 0.000138 | 9.231 | 0.00285 |
| PIMA | L2 | 0.228212 | 0.000958 | 20.113 | 0.000003448 |
| Adult | Mahalanobis | 4.236288 | 0.002453 | 3.0464 | 0.000821 |
| HELOC | Mahalanobis | 1.2506612 | 0.022587 | 0.12582 | 0.000949 |

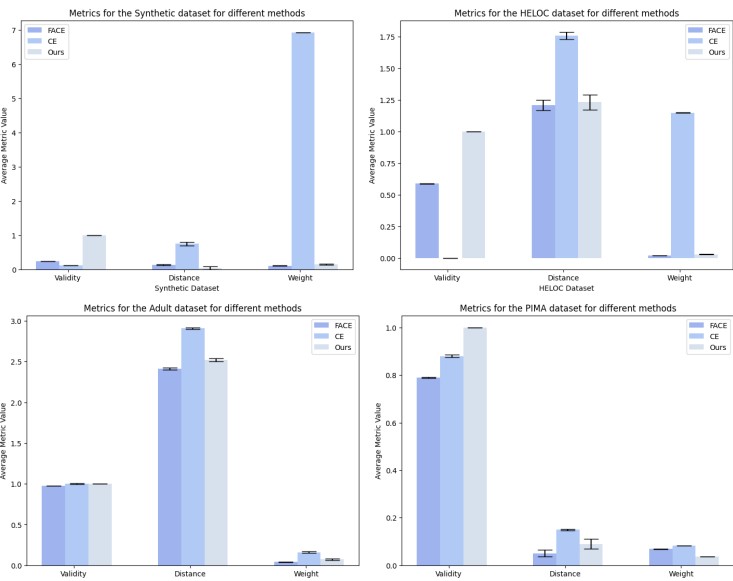

Figure 2: Evaluation of recourse paths using our method, FACE, and CE. Higher validity is better and lower distances and weights is better. Our method finds feasible (low distance and weights) recourse for all (validity=1) while other methods cannot.

**Results from augmentation:** Recourse paths are generated for 50 random samples for all four datasets. Figure 2 reports the metrics for our method on the four datasets, and we compare against two baselines: FACE, and nearest neighbor counterfactual explanations (CE). For FACE, a graph is constructed for each of the datasets with the distance function and threshold chosen for the graph as the same distance function and threshold that is learned using our method. The nearest neighbor counterfactual is the closest point to the point that achieves a positive outcome, and the path between these two points is evaluated. The choice of $\lambda$ for each dataset is provided in the Appendix A.1.Example recourse paths are provided in the appendix.

We note that across all datasets, both FACE and CE are unable to find valid paths for a set of points. This issue is especially exacerbated in the synthetic dataset and the HELOC dataset, hence these methods would not be able to provide recourse to all. Instead, our method always finds a valid path to recourse for every individual i.e., $VAL = 1$. The average distance and weight (lower is better) are consistently worse for the CE method. Our average distance and weight, both of which can be interpreted as measures of how "easy" the transitions in the path are, are comparable to FACE across datasets, demonstrating that the augmentation does not lead to substantially costlier paths.[3]

**Variation in $\lambda$:** The value of $\lambda$ determines our algorithm's convergence. A larger value corresponds to more steps but with lower average distance and weight between steps. This is demonstrated in figure 3, where we report 3 values of $\lambda$ for PIMA and the synthetic data. We report the evaluation metrics, and additionally report the average path lengths and runtime, since $\lambda$ directly impacts these measures. Choosing a larger $\lambda$ clearly leads to paths that are more "easy" (lower weights and lower distance), but this comes at the cost of a large path length and computation runtime. For PIMA a large lambda resulted in no recourse path for a few individuals (shown by validity being less than 1),

---

[3]The issue of not finding recourse is exacerbated when a point is isolated in a sparse region of the data. See Appendix A.7 for an analysis of path validity by the FACE method as a function of data sparsity.

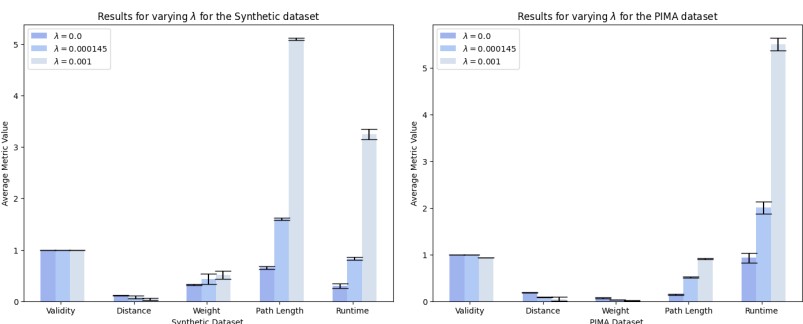

Figure 3: Evaluation of paths using different $\lambda$-s. The path length for the synthetic data has been scaled down by a factor of 10 and the weights have been scaled up by a factor of 10, for readability.

but this is because we had to kill the execution since augmentation was happening in very small steps (slow convergence). We highlight this to reflect on the need for an appropriate $\lambda$.

## 4 DISCUSSION

**On the availability and construction of feasible transitions data and choice of constraints:**
Ideally, every end-user subject to a models decision should be able to provide constraints under which they can obtain recourse, and existing methods such as (Sharma et al., 2020; Ustun et al., 2019) can include these in their optimization to find realistic recourse. However, having constraints for every individual, especially in large-scale applications is challenging. While we show that our framework is also applicable when provided a causal graph, often having access to complete or imperfect causal knowledge (Karimi et al., 2020) is also not viable, especially for large scale datasets.

Instead, our paper takes a data-driven approach where constraints for the path (to learn the distance function and threshold) can be provided based on domain knowledge by practitioners (eg., in finance and healthcare). For example, individuals can only increase their income by ten percent of their current income. While other methods that provide algorithmic recourse, several of which are implemented in (Pawelczyk et al., 2021). can accommodate for these constraints, certain constraints can lead to all these methods returning no recourse (eg., allowing income to only increase by ten percent might not provide recourse to low income individuals). Instead, our method still allows for multi-step recourse paths through augmentation, ensuring that a recourse path is provided.

**On the choice of distance functions for algorithmic recourse:** Seminal methods for counterfactual explanations (Wachter et al., 2017; Ustun et al., 2019; Sharma et al., 2020; Mothilal et al., 2020) and several others implemented in (Pawelczyk et al., 2021) use some $l_p$ norm to generate counterfactual explanations. Hence, we consider the L1 and L2 distance functions. (Chen et al., 2020) show that using the Mahalanonbis distance can capture feature interactions. Cosine and Jensen-Shannon distances are not widely used in the recourse literature, however, they show that distance functions do not need to be metric for our method to work. Other distance functions can also easily be incorporated into our proposed method for distance function and threshold learning.

## 5 CONCLUSION

We studied the problem of providing actionable recourse by suggesting multi-step transitions (recourse paths) to individuals. We presented an augmentation algorithm that empirically provides recourse for all. To strengthen the feasibility of recourse, we are the first to study the problem of PAC learning of the ground-truth transitions through a hypothesis class of distance functions and thresholds. We see two main limitations in our work which are opportunities for future work: 1) for the hypothesis class in (1), we can studying more expressive feasibility functions, e.g., feasibility measures using casual constraints and 2) learning feasible transitions involved label generation ; future directions will include using human annotators for labeling transitions.

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

# A  APPENDIX

## A.1  DETAILS ON DATASETS AND MODELS

Table 2 provides datasets and models used, along with optimization parameters used to generate the results in 2. All datasets have been randomly split into train and test in a 75:25 ratio.

## A.2  DETAILS ON BAYESIAN OPTIMIZATION

Bayesian optimization works by constructing a posterior distribution of functions (gaussian process) that best describes the function you want to optimize. As the number of observations grows, the posterior distribution improves, and the algorithm becomes more certain of which regions in parameter space are worth exploring and which are not. Details on the method and implementation can be found here Nogueira (2014–).

The empirical convergence of our method depends on the chosen value of $\lambda$ (as shown in theorem 2.8). In general, we observe that larger learned distance thresholds (Adult and HELOC) result in larger optimal $\lambda$. The tried $\lambda$ values range between 0 and 0.2. Beyond those values, all methods do not converge to a solution due to very small steps towards the decision boundary. The number of iterations and number of initial points for bayesian optimization are varied from 10-100 in steps of 10, and the value that returns the lowest distance average.

Table 2: Details on datasets, models, and parameters

|  | Samples | Train Accuracy % | Test Accuracy % | B.O. initial points | B.O. iterations | $\lambda$ |
|---|---|---|---|---|---|---|
| Synthetic Data | 100 | 96.67 | 96.48 | 10 | 10 | 0.000145 |
| PIMA | 768 | 76.12 | 75.23 | 50 | 50 | 0.000145 |
| Adult | 42556 | 81.20 | 81.04 | 100 | 10 | 0.1 |
| HELOC | 3614 | 71.39 | 71.59 | 50 | 50 | 0.1 |

## A.3  SYNTHETIC CAUSAL DATA GENERATION

We experimented on causal data to test the algorithm which learns the distance threshold $\tau$ and the regularization parameter $\lambda$ from the feasibility labels. Below, we describe the data generating process.

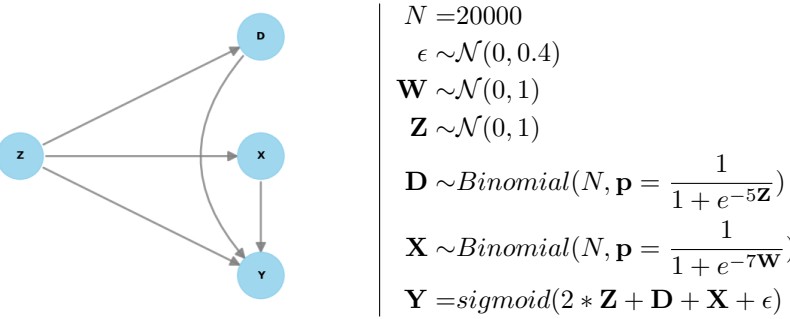

$$N = 20000$$
$$\epsilon \sim \mathcal{N}(0, 0.4)$$
$$\mathbf{W} \sim \mathcal{N}(0, 1)$$
$$\mathbf{Z} \sim \mathcal{N}(0, 1)$$
$$\mathbf{D} \sim Binomial(N, \mathbf{p} = \frac{1}{1 + e^{-5\mathbf{Z}}})$$
$$\mathbf{X} \sim Binomial(N, \mathbf{p} = \frac{1}{1 + e^{-7\mathbf{W}}})$$
$$\mathbf{Y} = sigmoid(2 * \mathbf{Z} + \mathbf{D} + \mathbf{X} + \epsilon)$$

Figure 4: Directed Acyclic Graph (DAG) and the structural equation model used to generate the distribution of 'origin points' in our synthetic causal dataset.

First, we generate $N = 20,000$ samples according to the data-generating process described in Figure 4, to obtain $\Omega = \{(X^i_{origin}, D^i_{origin}, Z^i_{origin}, Y^i_{origin})\}^N_{i=1}$.

Then, we define feasible and infeasible transitions $f(X_{origin}, D_{origin}, X_{target}, D_{target}) \in \{1, 0\}$ using monotonicity constraints on $X$ and $D$:

$$f(X_{origin}, D_{origin}, X_{target}, D_{target}) = \begin{cases} 1 \text{ if } X_{target} >= X_{origin} \text{ and } D_{target} <= D_{origin} \\ 0 \text{ otherwise} \end{cases}$$

We then proceed, for each initial point, to sample $n_s = 5$ feasible and $n_s = 5$ infeasible points from the counterfactual distirbution under interventions we defined as feasible and infeasible. Upon sampling 5 values of (X,D,Z,Y) corresponding to feasible interventions, we impose an additional constraint to further re-brand these cases: we will call them feasible if in addition to satisfying the monotonicity constraints defined by $f$, the resulting change in $Z$ is not too large.[4] The process is described in Algorithm 3.

---

**Algorithm 3** Synthetic Data Feasibility Labeling Algorithm

---

**Input:** Set $\Omega$, function $f$, parameters $\tau_f, n_s$.
1: **for** each data point $\omega \in \Omega$ **do**
2:    **for** feasibility label $\varphi \in \{0, 1\}$ **do**
3:       $n_{sampled} \leftarrow 0$
4:       **while** $n_{sampled} < n_s$ **do**
5:          Sample $(X_{target}, D_{target})$ s.t. $f(X_{origin}, D_{origin}, X_{target}, D_{target}) = \varphi$
6:          $(X^i_{target}, D^i_{target}, Z^i_{target}, Y^i_{target}) \sim P_{do(X \to X_{target}, D \to D_{target})}(X, D, Z, Y)$
7:          $n_{sampled} \leftarrow n_{sampled} + 1$
8:          **if** $\varphi == 1$ {*for 'feasible' labels*} **then**
9:             $\tilde{\varphi} \leftarrow \begin{cases} 1 \text{ if } |Z_{target} - Z_{origin}| < \tau_f \\ 0 \text{ otherwise} \end{cases}$
10:         **end if**
11:       **end while**
12:    **end for**
13: **end for**

---

For step 6 of the Algorithm 3, we used the `dowhy` (Sharma & Kiciman, 2020) package fitted to the DAG in Figure 4. As the value of parameter that controls the additional constraint on feaisbility labels in Step 8, we used $\tau_f = 0.1$

### A.4 PROOFS FOR THEOREMS

***Proof of Theorem 2.2.*** Consider any set of $N$ pairs of individuals $p_i = (x^i, y^i) \in \mathcal{I}^2$, where $i \in [N]$. Any "distance" function $d : \mathcal{I}^2 \mapsto \mathbb{R}_{\geq 0}$ explicitly induces an ordering $\pi_d(1), \pi_d(2), \ldots, \pi_d(N)$ of $[N]$, such that $d(p_{\pi_d(1)}) \leq d(p_{\pi_d(2)}) \leq \ldots \leq d(p_{\pi_d(N)})$; by assuming some infinitesimal noise on $d$ all ties are broken and the ordering is unique. The functions $d \in \mathcal{D}$ can produce at most $|\mathcal{D}|$ such pair orderings. For each ordering produced by a $d \in \mathcal{D}$, you can have $N + 1$ different labelings depending on the chosen threshold $\tau$. Hence, at most $|\mathcal{D}|(N + 1)$ different labelings that can be produced by the hypotheses of $\mathcal{H}$. On the other hand, the total number of labelings is $2^{N+1}$. Given that $|\mathcal{D}| < \frac{2^N}{N+1}$, there must be at least one labeling that cannot be produced. Therefore, $VC \leq N$, since any set of $N$ pairs cannot be shattered. $\square$

***Proof of Theorem 2.3.*** For a given $d \in \mathcal{D}$, order and re-index all $(x^i, y^i, h^*(x^i, y^i)) \in S$ such that $d(x^1, y^1) \leq d(x^2, y^2) \leq \ldots \leq d(x^m, y^m)$. It is clear that trying a threshold that is between two consecutive distances in the above ordering will not affect the empirical error, and therefore it suffices to only focus on the $d(x^i, y^i)$ as values for the threshold $\tau$. $\square$

***Proof of Theorem 2.5.*** Consider any set of $2n + 1$ pairs of individuals $p_i = (x^i, y^i) \in \mathcal{I}^2$, where $i \in [2n + 1]$. Assume w.l.o.g. that $\mathcal{H}$ can shatter the set of pairs and can produce all possible labelings. For notational convenience, we use $c^i_j = f_j(x^i_j, y^i_j)$. At first, for each feature $j \in [n]$, we are interested in the (potentially) two pairs $i^j_{max}, i^j_{min} \in [n]$ such that $i^j_{max} = \arg\max_{i \in [n]} c^i_j$ and $i^j_{min} = \arg\min_{i \in [n]} c^i_j$. Let $M = \{i \in [2n + 1] \mid \exists j \in [n] \text{ with } i = i^j_{max} \vee i = i^j_{min}\}$. Clearly, $|M| \leq 2n$, and hence by the pigeonhole principle there exists $i^* \in [2n + 1]$ such that $i^* \notin M$.

---

[4]While using the *do-* calculus can be thought of as requiring that a resulting data point is *plausible*, this second-order feasibility constraint on the feature $Z$ can be understood as additionally requiring that the change is *not too costly*.

Now consider a labeling $\ell : [2n+1] \mapsto \{0,1\}$, such that $\ell(p_{i^*}) = 1$ and $\ell(p_i) = 0$ for all $i \in [2n+1] \setminus \{i^*\}$. Regardless of the chosen threshold, $\mathcal{H}$ can only produce this labeling if there's a $\beta_\ell \in \mathbb{R}^n_{\geq 0}$ such that $d_{\beta_\ell}(p_{i^*}) < d_{\beta_\ell}(p_i)$ for all $i \in [2n+1] \setminus \{i^*\}$. For that to be true, since $g()$ is strictly increasing, the following $2n$ inequalities must hold:

$$\sum_{j=1}^{n}(c_j^{i^*} - c_j^i) \cdot \beta_{l,j} < 0, \quad \forall i \in [2n+1] \setminus \{i^*\} \tag{3}$$

Since $i^* \notin M$, for each $j \in [n]$, there exists at least one coefficient $c_j^{i^*} - c_j^i > 0$ and at least one coefficient $c_j^{i^*} - c_j^i < 0$. This means that in a Gaussian elimination processes applied only to the coefficients of the system, we can only use positive multiplicative factors that will not be altering the directions of the inequalities. Therefore, for any feature $\bar{j}$ that we might choose, we will eventually end up with an inequality of the form $C_{\bar{j}} \cdot \beta_{\ell,\bar{j}} < 0$. The constant $C_{\bar{j}}$ is the one resulting from the Gaussian elimination process. Since, $\beta_{\ell,\bar{j}} \geq 0$ we thus have $C_{\bar{j}} < 0$.

Now consider a labeling $\ell' : [2n+1] \mapsto \{0,1\}$, such that $\ell'(p_{i^*}) = 0$ and $\ell'(p_i) = 1$ for all $i \in [2n+1] \setminus \{i^*\}$. Regardless of the chosen threshold, $\mathcal{H}$ can only produce this labeling if there's a $\beta_{\ell'} \in \mathbb{R}^n_{\geq 0}$ such that $d_{\beta_{\ell'}}(p_{i^*}) > d_{\beta_{\ell'}}(p_i)$ for all $i \in [2n+1] \setminus \{i^*\}$. For that to be true, since $g()$ is strictly increasing, the following $2n$ inequalities must hold:

$$\sum_{j=1}^{n}(c_j^i - c_j^{i^*}) \cdot \beta_{\ell',j} < 0, \quad \forall i \in [2n+1] \setminus \{i^*\} \tag{4}$$

Using the exact same analysis that led to $C_{\bar{j}} < 0$, we can see that in this case we will have $C_{\bar{j}} > 0$. Therefore, $\mathcal{H}$ cannot simultaneously achieve both of the labelings $\ell, \ell'$. $\qquad \square$

***Proof of Theorem** 2.6.* Consider any given set $S$ that contains $m$ pairs $(x^i, y^i)$, with $i \in [m]$. Computing an ERM classifier corresponds to finding $\beta$ and $\gamma$ that satisfy as many of the following $m$ constraints as possible:

$$\sum_{j=1}^{n} \beta_j \cdot f_j(x_j^i, y_j^i) \leq \gamma, \quad \text{for } i \in [m]$$

This is because $g$ is monotonically increasing and we can set $\tau = g(\gamma)$. Since, the values $f_j(x_j^i, y_j^i)$ are given constants, this is a linear system. Even though this problem is NP-hard (reduction to MAX-SAT) there are efficient and very accurate $O(1)$-approximation algorithms for it (Williamson & Shmoys, 2011) (best ratio achieved by an SDP approach and is 0.7846). $\qquad \square$

***Proof of Theorem** 2.8.* We claim that under the assumption of the theorem statement, the $f()$ value of all points along the path will be strictly increasing. We show this via induction.

- Induction Basis: For the second point of the path, the $f()$ value will increase because will apply the theorem hypothesis to the starting point of the path.

- Inductive step: Say that for our path so far the $f()$ value is increasing. Consider the end point $x$. For $x$ there exists $y$ such that $f(y) - f(x) > \frac{\lambda}{\min_{a,b} w(a,b)}$. We now claim that $y$ cannot be in the path. If it is, the path contains a point with a $f()$ value strictly larger than that of $x$. Therefore, since $y$ is not in the path, the optimization problem will necessarily pick $y'$ with $f(y') - f(x) > 0$; if not we have $f(y') - f(x) + \frac{\lambda}{w(y',x)} < \frac{\lambda}{\min_{a,b} w(a,b)} < f(y) - f(x) + \frac{\lambda}{\min_{a,b} w(a,b)}$. $\qquad \square$

## A.5 EXPERIMENTS ON XGBOOST AND NEURAL NETWORK

Experiments are also performed on XGBoost and Neural network models for the PIMA Diabetes dataset. The results are shown in 5. We observe a similar trend as in the results for the logistic regression model.

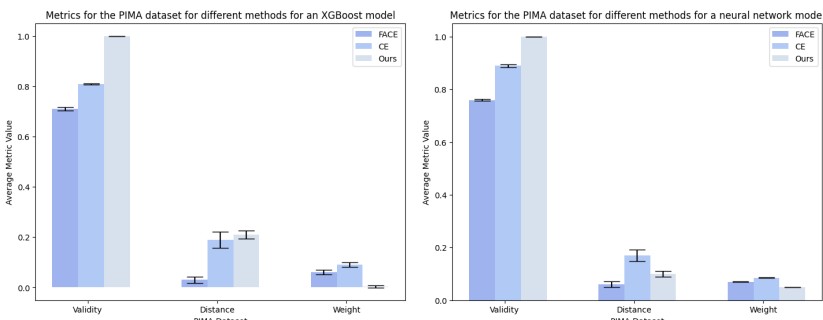

Figure 5: Results on XGBoost and Neural network models for the PIMA Diabetes dataset

Table 3: Recourse path for a query in the PIMA dataset, compared to the CE method

|  | Glucose | BloodPressure | Insulin | BMI |
|---|---|---|---|---|
| Nearest Counterfactual | 159 | 66 | 0 | 30.4 |
|  | 151 | 60 | 0 | 26.1 |
| Ours | 159 | 66 | 0 | 30.4 |
|  | 157.82 | 65.131 | 0 | 30.2 |
|  | 155.586 | 64.84 | 0 | 30.1 |
|  | 153.40 | 64.11 | 0 | 29.32 |
|  | 151.20 | 64.60 | 0 | 29.32 |
|  | 151.20 | 64.20 | 0 | 29.31 |
|  | 149.26 | 65.20 | 0 | 29.30 |

## A.6 ADDITIONAL EXAMPLE RECOURSE PATH

Table 3 shows an example recourse path for PIMA. The first row in the table for both nearest counterfactual and ours is the input point for which we are seeking to provide the recourse path. FACE is unable to find a path for this individual. CE suggests a single step recourse path with drastic changes in the values of each feature except insulin. Instead, our method provides a sequence of smoother steps that lead to the positive outcome.

An additional example for a recourse path for the HELOC dataset is shown in Table 4.

Table 4: Recourse path for a query in HELOC comparing CE with our method. Here $P(Y = 1)$ refers to the probability of positive (desirable) classification, with scores corresponding to loan denial highlighted in red, and approval qualifying scores in green. The feature names are abbreviared as follows: ERE : ExternalRiskEstimate ; MSOTO : MSinceOldestTradeOpen ; AF : AverageMInFile ; NFRB : NetFractionRevolvingBurden; NFIB : NetFractionInstallBurden

|  | ERE | MSOTO | AF | NFRB | NFIB | $P(Y = 1)$ |
|---|---|---|---|---|---|---|
| Original point | 68. | 124. | 51. | 72. | 73. | 0.26 |
| Nearest Counterfactual | 75. | 161. | 67. | 40. | 75. | 0.51 |
| Our path | 68. | 124. | 51. | 72. | 73. | 0.26 |
|  | 75.7 | 152.7 | 77.4 | 64.3 | 62.8 | 0.48 |
|  | 83.4 | 181.3 | 103.9 | 56.7 | 52.7 | 0.71 |

## A.7 ANALYSIS OF VALIDITY FOR THE BENCHMARK METHOD (FACE)

Central to our contribution is the argument that, unlike our approach which guarantees recourse, salient path-based algorithms may fail to provide recourse for some negatively classified data instances. We therefore examine the relationship between dataset sparsity and the rate of successful generation of a recourse path using FACE.

Figure 6 below demonstrates this relationship on the HELOC data. We sample $k$ negatively classified data instances from the training set. For each sample of $k$ points, we run FACE algorithm and record the percentage of instances for which FACE successfully identified a recourse path (all other instances receiving no recourse). We further compute average pairwise Mahalanobis distance for the sampled $k$

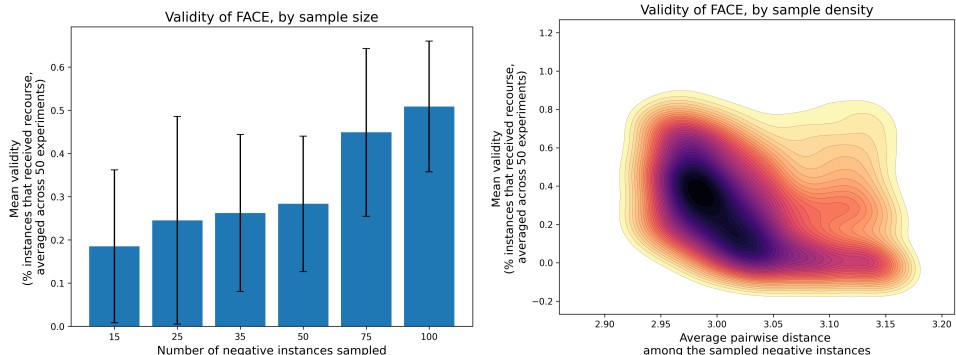

Figure 6: Validity (i.e., success rate) of the FACE algorithm on subsets of the HELOC dataset. Left: as a function of the number of negative instances selected into the sample. Right: as a function of the density of the sampled negative instances set. For points belonging to the sparse regions of the data, FACE may not identify a recourse path.

points as a proxy for sample density. We repeat the experiment 50 times for each $k$. Additionally, we exclude the successful paths of length two (consisting only of the initial point and its counterfactual), such paths representing 'lucky draws' of negative instances that are already very close to the decision boundary. The graphs report results averaged over 50 experiments.

It is important to note that even when the set of negative instances in the full dataset is dense, FACE may fail to identify a recourse path for instances which are isolated in a *sparse local region* of the data manifold. This is especially important since such sparse regions may represent individuals who are underrepresented in the dataset, which in turn might be associated with membership in protected social groups. It is known that sampling bias adversely affects various demographic groups Kokel et al. (2023); Sonoda (2023). If membership in such a group is also associated with being in a sparse region of the data manifold, failure of path-based methods might disproportionally affect members of such groups.