# OpenReview forum: "Everyone Deserves Recourse: Feasible Recourse Paths Using Data Augmentation"
_ICLR.cc/2025/Conference — ICLR 2025 Conference Withdrawn Submission_

### Official Review · Reviewer_r6Vx · 2024-11-04

**Soundness:** 1
**Presentation:** 1
**Contribution:** 1
**Rating:** 3
**Confidence:** 4

**Summary:**

This paper addresses limitations in the FACE algorithm, which is used to provide feasible recourse by finding a path between factual and counterfactual instances through training data points. FACE's design assumes that the distance between steps is manageable for end-users, yet it fails to find recourse if this distance exceeds a specified threshold. In response, the authors propose augmenting the training data with artificial samples to bridge gaps between sparse regions in the input space, enhancing the likelihood of finding recourse. They also introduce a mechanism to learn a suitable distance function and threshold parameter, controlling the step margin in the recourse path. However, some theoretical aspects and practical relevance of the proposed method remain unclear.

**Strengths:**

- **Improving on FACE's Limitations**: The paper identifies and addresses an issue in the FACE algorithm: its tendency to terminate prematurely when there are significant distances between feasible training data points on the recourse path, potentially leaving users without recourse options.
- **Increased Accessibility of Recourse**: By augmenting the dataset with synthetic samples, the proposed method aims to provide more universally accessible recourse, addressing the paper’s central argument that recourse should be available to all individuals

**Weaknesses:**

- **Privacy Concerns with FACE as a Basis**: Using FACE as the foundational algorithm raises significant privacy concerns, as the method inherently relies on traversing data points from other users, which exposes sensitive information of other unsuspecting users. This risk, highlighted in recent studies (e.g., [1]), casts doubt on the real-world applicability of FACE-based recourse due to its clear privacy violations.

- **Limited Evaluation Scope**: The empirical evaluation lacks breadth and depth. The comparison is limited to two other recourse algorithms, neglecting the diverse set of algorithms in the CARLA library [2] that could provide a more thorough assessment of the proposed method's effectiveness. For example, any gradient based algorithm can be return a recourse path. Hence, the algorithms in [2] serve as natural baselines for comparison. Additionally, the core evaluation is restricted to a logistic regression model. Since the authors claim the method is model-agnostic, it would be valuable to test all datasets on other model types, such as neural networks and tree-based classifiers, to validate its general applicability.

- **Inadequate Handling of Categorical Variables**: The paper does not sufficiently address how categorical variables are managed, which is crucial in tabular datasets where they are common. Without clarification, it remains unclear whether the proposed approach is suited for real-world applications that often involve categorical features.

- **Not Useful and Disconnected Theoretical Contributions**: The theoretical results in Section 3 are not well-integrated with the empirical findings. While they cover basic insights, such as bounding the VC dimension of a specific hypothesis class, these insights don’t directly support the method's empirical performance. Additionally, the feasibility learning appears overly complex without practical benefits, especially given the lack of efficient algorithms for managing a large set of parameterized distance functions. Fruther, the paper does not provide theoretical guarantees that the synthetic data augmentation will remain within the data manifold. Such guarantees are necessary to ensure that the generated samples represent realistic and plausible data points, enhancing the method's reliability and applicability.

-----
**References**

[1] Pawelczyk et al (2023), On the Privacy Risks of Algorithmic Recourse, AISTATS 2023, https://arxiv.org/abs/2211.05427

[2]  Pawelczyk et al (2021), Carla: a python library to benchmark algorithmic recourse and counterfactual explanation algorithms, NeurIPS 2021, https://arxiv.org/abs/2108.00783

**Questions:**

- Can the authors explain why the compare two different starting points in Figure 1? For FACE, you use one starting point, and for your algorithm you are using another one? This seems very inconsistent and and does not make these two scenarios comparable.
- Can the authors provide specific details on how their method handles categorical variables? What the approach for encoding categorical features and what's the method's performance on a dataset with a mix of numerical and categorical variables?
- How do the VC dimension bounds relate to the sample complexity in the experiments? How do the theoretical guarantees inform the practical implementation of the method?

---

### Official Review · Reviewer_6FSG · 2024-11-04

**Soundness:** 1
**Presentation:** 2
**Contribution:** 2
**Rating:** 3
**Confidence:** 4

**Summary:**

This paper tackles the problem of providing paths in algorithmic recourse. Unlike the previous work which assume fixed cost and threshold functions, this paper attempts to instead search for both of these quantities. They also include a data augmentation strategy to construct paths for individuals which have sparse neighbourhoods.

**Strengths:**

- The paper builds up on existing recourse literature and provide a solution for otherwise neglected issue.
- The problem statement is well motivated
- Provides several theoretical results regards to learning feasible transitions

**Weaknesses:**

- **Formulation**: The paper doesn’t motivate why searching for distance functions helps, in particular because initial $h^*$ is obtained by assuming distance function $ell_1$ as described in lines 381-388
- **Writing:** In line 389, “”learning $d,\tau$”, there is no mention of $\tau$.Crucial Details on how transition weights are created is missing(line 395).
- **Limited Experimental Results:** Authors do not compare their methods to any single step algorithm other than nearest neighbours, while several papers on path based recourse are mentioned in related works sections.

**Questions:**

- Is it possible to add comparisons of cost between the starting and the end point? Does path-based approaches lead to higher cost than single step methods?
- Is there any justification for creating the labelled dataset using one cost function and then trying to learn it?
- Is it possible to include more comparisons with more recent methods, as mentioned in related work section?
- Can this paper be considered extension of work such as Trustworthy Actionable Perturbations(Friedbaum et al., 2024)

[1] Jesse Friedbaum, Sudarshan Adiga, and Ravi Tandon. Trustworthy actionable perturbations, 2024a. URL https://arxiv.org/abs/2405.11195

Other comments:
Line 244: formatting error in $\rho$-erm definition

---

### Official Review · Reviewer_bYZM · 2024-11-06

**Soundness:** 2
**Presentation:** 2
**Contribution:** 2
**Rating:** 3
**Confidence:** 3

**Summary:**

This paper considers the path-based recourse: instead of viewing recourse as a terminal node, this paper considers the whole path from the original input to the terminal node. I agree that a path-based is more friendly because it provides a step-by-step guidance to flip the prediction of the algorithm. Path-based recourses relies on the construction of a graph. The specific problem considered in this paper is that the graph may not be connected. As a consequence, there may not exist a path to reach the positive region.

To solve the problem, the paper considers augmenting data: adding new nodes with connecting edges to the graph. By doing this, the paper guarantees that every input has a feasible path.

**Strengths:**

The most interesting thing I found about this paper is the application of theoretical machine learning into the recourse problem. This is definitely something new that I have not seen in previous papers on recourse generation

**Weaknesses:**

1. Recourse problem is a human-centric problem. The paper lacks the human-understanding of why the graph is not connected. Simply augmenting points to make the graph connected could not solve the root cause of the problem.

2. The augmented nodes lacks interpretability. There is no argument to support that the new nodes correspond to feasible features in the real world. Thus, one could not guarantee that the proposed path is realistic.

**Questions:**

1. Could the author explain around line 343 why the second problem ``might give very realistic feature profiles”?

2. How is the Mahalanobis distance defined in the experiment?

3. Should $\tau$ be tuned or should be learnt. In my opinion, I think $\tau$ should be tuned: it depicts the maximal efforts that the model requires in each transition. It is unclear to me why it should be learned in this paper. For an input without a path, then the model can simply inform the person that the effort threshold should be increased in order to obtained a feasible path. The model can also recommend different paths for different levels of $\tau$, which can totally resolve the unconnectedness problem.

4. The order that we pick $x \in N_p$ may affect the final set of $U_n$. Thus, a sequential approach to augment the dataset is not optimal. Could the authors devise a holistic method that takes a given graph, and add a minimal number of additional nodes such that the graph becomes connected?

---

### Note · Authors · 2024-11-25

I have read and agree with the venue's withdrawal policy on behalf of myself and my co-authors.